# An ABCG-Type Transporter Facilitates ABA Influx and Regulates Camptothecin Biosynthesis in *Camptotheca acuminata*

**DOI:** 10.3390/ijms232416120

**Published:** 2022-12-17

**Authors:** Yanyan Wang, Yang Wang, Hefei Bai, Yuqian Han, Fang Yu

**Affiliations:** 1School of Biological Engineering, Dalian Polytechnic University, Dalian 116034, China; 2WX Biologics, Wuxi 214000, China

**Keywords:** *Camptotheca acuminata*, camptothecin biosynthesis, transport, ABA transporter

## Abstract

Camptothecin (CPT) and its derivatives from *Camptotheca acuminata* have antitumor effects as a DNA topoisomerase I inhibitor. Previous studies have shown that application of exogenous abscisic acid (ABA) significantly promoted the accumulation level of CPT and induced the expression of CPT biosynthetic genes, which revealed that ABA signaling is effectively involved in regulating CPT biosynthesis in *C. acuminata*. In this study, an ABA transporter, *CaABAT*, which encodes a plasma membrane protein belonging to the ABCG subfamily, was identified in *C. acuminata*, and its ABA import activity was confirmed by transport assay in yeast cells. Real-time PCR analysis showed that CaABAT was predominately expressed in *C. acuminata* leaves and its expression could be significantly upregulated by exogenous ABA treatment. Silencing of *CaABAT* down-regulated the expression of ABA response genes, which indicated that translocation of ABA by CaABAT should initiate changes in plant physiological status in response to ABA signaling, thus leading to decreased expression of CPT biosynthesis pathway genes and low accumulation levels of CPT in *C. acuminata*.

## 1. Introduction

Camptothecin (CPT), a monoterpene indole alkaloid (MIA), was initially isolated and identified from *Camptotheca acuminata*. Later investigation showed that CPT and its derivatives could inhibit DNA topoisomerase I activity, which led to it being used as a promising anti-tumor drug for treating cancer diseases in clinical therapy [1,2,3]. However, similarly to other valuable specialized metabolites in plants, low accumulation of CPT in *C. acuminata* limits its supply in the market [4]. In order to promote the production of CPT in plants, many strategies have been applied, such as the overexpression of pathway genes, and application of elicitors and/or precursor compounds in suspension cells and/or hairy root cultures. Some plant hormones, such as jasmonic acid (JA) [5,6], gibberellic acid (GA) [7], salicylic acid (SA) [8], and abscisic acid (ABA) [9], have already been identified to be involved in regulating plant specialized metabolite biosynthesis as effective elicitors, and many previous reports also revealed that application of these plant hormones could successfully induce the expression of CPT biosynthesis genes, thus promoting CPT accumulation in *C. acuminata* [5,6,7,8,9,10].

ABA, as the most effective elicitor for promoting CPT biosynthesis according to a previous report [10], is principally biosynthesized in vascular tissues and then transported to the target cells (such as the guard cells) through specific transporters to achieve its biological functions [10,11,12]. In *Arabidopsis thaliana*, several ABA transporters have been identified, such as AtABCG25, AtABCG31, AtABCG30, AtABCG40, AIT1/NRT1.2, DTX50, and NPF5.1 [13,14,15,16,17].

Among these ABA transporters, AtABCG40 is localized at the plasma membrane and functions as an ABA importer for importing ABA into guard cells [18], where CPT could be effectively synthesized in *C. acuminata* [19]. Plants lacking functional AtABCG40 exhibit less sensitivity to ABA, including reduced expression of ABA-responsive genes and impaired stomatal closure [18].

In the present study, a homolog of *AtABCG40*, *CaABAT*, was isolated from *C. acuminata*. Its ABA transport activity was determined in yeast cells and its effects on CPT biosynthesis were also investigated in *C. acuminata* plants. Our results strongly suggest that regulation of plant hormone translocation by certain specific transporters could be a feasible strategy for effective production of valuable plant specialized metabolites.

## 2. Results

### 2.1. Isolation and Characterization of CaABAT

Sequencing results indicate that *CaABAT* contains 4374 bp of coding region, and it encodes 1458 amino acids with 168.12 kDa. For analyzing conserved domains of *CaABAT*, the amino acid sequences of *CaABAT* with AtABCG40 were aligned using BLAST and InterPro. The results showed that the homology of the two sequences reached 70%, and the homologous proteins contained specific domains for transporters, an ABC transporter domain (81–143 aa), ATP binding cassette transporter PDR-like subfamily G domain 1 (152–425 aa) and domain 2 (857–1112 aa), ABC-2-type transporter (504–716 and 1185–1399 aa), plant PDR ABC transporter-associated (721–784 aa), and ATPase domain (177–409 and 897–1089 aa) (Figure 1A). Furthermore, phylogenetic analysis of *CaABAT* with other ABA transporters was performed (Figure 1B), and a prediction of the secondary structure suggested that the protein had six transmembrane structures outside and inside the bilayer region (Figure 1C and Appendix A). These results demonstrate that *CaABAT* encodes a plasma membrane protein and belongs to the ABCG subfamily.

### 2.2. Functional Identification of CaABAT in Yeast Cells

To examine whether CaABAT functions as an ABA transporter, CaABAT was expressed in the yeast strain AD12345678, which lacks eight major yeast ABC transporter genes [20]. The results showed that ABA could inhibit yeast growth, and this inhibition was more obvious by overexpressing *CaABAT* in yeast cells, which indicated that CaABAT might have ABA import activity, thus increasing the accumulation level of ABA in cells to raise the inhibitory effect of ABA on yeast growth (Figure 2A). To further confirm ABA transport activity of CaABAT, a transport assay was carried out by analyzing uptake amounts in yeast cells with overexpressing *CaABAT* compared to that with overexpressing empty vector. Results showed that the accumulation levels of ABA were much higher in yeast cells by overexpressing *CaABAT* in comparison with empty vector control, which validated the ABA import activity of CaABAT in yeast cells (Figure 2B).

### 2.3. Expression Analysis and Subcellular Localization of CaABAT

For analyzing the expression profile of CaABAT, three tissues (root, stem, and leaf) from *C. acuminata* plant were collected for gene expression analysis by real-time PCR. Results showed that *CaABAT* was predominately expressed in leaves, while the relative expression levels of *CaABAT* in root and stem were about 10 times lower than that in leaves (Figure 3A). Furthermore, an ABA responsive gene [21], *CaRD29B*, which could be induced by ABA treatment, was also investigated for its expression profile in *C. acuminata* plant. Results showed that the highest expression level of *CaRD29B* was observed in leaves, which was similar to the expression profile of *CaABAT*. These similar expression profiles of *CaABAT* and *CaRD29B* imply that CaABAT might participate in ABA signaling transduction in response to abiotic stresses by translocating ABA to trigger increased expression of CPT biosynthesis genes for promoting CPT accumulation in *C. acuminata* leaves (Figure 3B).

To determine the subcellular localization of CaABAT, both pBIGD-*GFP* and pBIGD-*GFP-CaABAT* vectors were transformed into *Agrobacterium* for treating onion epidermal cells. GFP signals were detected by a confocal microscope, and results showed that GFP-CaABAT fusion protein localized at plasma membrane in onion epidermal cells (Figure 3C). Considering the ABA import activity of CaABAT in yeast cells, CaABAT should function as an ABA importer in *C. acuminata*.

### 2.4. Exogenous Plant Hormone Treatments on C. acuminata Seedlings

Exogenous ABA was applied to treat *C. acuminata* seedlings for analyzing the expression of *CaABAT* and CPT biosynthesis pathway genes in response to ABA treatments. The seedlings were treated with ABA for 2 and 4 h, separately, and the expression of *CaABAT* and *CaRD29B* could be successfully induced (Figure 4A), which indicated that exogenous ABA triggered a plant response to ABA signaling. For investigating the effect of exogenous ABA on the expression of CPT biosynthesis genes, six pathway genes were selected for the examination. Results showed that four pathway genes (*CaCYC1*, *CaG8O*, *Ca7DLS*, and *Ca7DLGT*) could be successfully induced by ABA treatment, in which the transcript level of *CaCYC1* increased about 50 times in response to exogenous ABA treatment in 4 h (Figure 4B). For analyzing the accumulation levels of CPT, seedlings were treated with ABA for 24, 48, and 72 h. It was shown that the highest accumulation level could be observed after 24 h treatment, and then decreased to original levels (Figure 4C).

Besides ABA, two other important plant hormones, MeJA and SA, were also selected for testing their effects on the expression of *CaABAT*. Results showed that both MeJA and SA had only slight effects for inducing the expression of *CaABAT*, while obvious inducing the expression of responding genes, *CaMYC2* and *CaNPR1,* could be observed (Appendix A). These gene expression profiles of *CaABAT* in responding to different plant hormone treatments demonstrated that CaABAT should be mainly involved in ABA signaling for regulating CPT biosynthesis in *C. acuminata*.

### 2.5. Silencing and Overexpression of CaABAT in C. acuminata

To further investigate the effect of *CaABAT* on the biosynthesis of CPT, the VIGS approach was carried out for silencing the expression of *CaABAT* in *C. acuminata* leaves (Figure 5). Results showed that the accumulation level of CPT decreased (by approximately 33%) in *CaABAT*-silenced leaves in comparison with that in empty vector control leaves (Figure 5B), while the expression levels of CPT biosynthesis pathway genes *(CaCYC1*, *CaG8O*, and *Ca7DLS*) were down-regulated in *CaABAT*-silenced leaves (Figure 5C). Silencing of *CaABAT* also resulted in decreased expression of ABA responding gene, *CaRD29B* (Figure 5A), which indicated that inefficient import of ABA mediated by *CaABAT* could affect translocation of ABA in cells, thus reducing CPT production by repressing ABA signaling in *C. acuminata* leaves. Interestingly, when *CaABAT* was overexpressed in *C. acuminata* leaves, no obvious variation in CPT accumulation levels could be observed between *CaABAT*-overexpressing and empty vector control leaves (Appendix A). Since the biosynthesis of CPT in *C. acuminata* reveals complicated cell-type specific compartmentation [19], overexpression of *CaABAT* in all cell types might not guarantee endogenous that ABA is efficiently moved into target cell types, where CPT is synthesized, for promoting CPT biosynthesis. Down-regulation of CPT biosynthesis genes and the ABA biosynthesis gene, *CaNCED,* and up-regulation of the ABA responding gene, *CaRD29B* (Appendix A), also verified cell-type specific compartmentation for CPT biosynthesis in *C. acuminata*.

## 3. Discussion

### 3.1. CaABAT Belongs to the ABCG Subfamily and Is Involved in ABA Transport

As an important plant stress hormone, ABA pervasively exists in plants. The level of endogenous ABA is a major determinant of ABA sensing that is maintained by ABA biosynthesis, catabolism, and transport. At present, ABA transporters from *Arabidopsis thaliana*, such as AtDTX50, AtABCG25, AtABCG40, AtABCG31, AtABCG30, AIT1, and NPF5.1, have been thoroughly studied [13,14,15,16,17,18]. These transporters belong to three protein families: the multidrug and toxin efflux transporter family, nitrate transporter 1/peptide transporter family, and ABCG family [13,14,15,16]. Among these ABA transporters, AtABCG40 was identified to be importing ABA into guard cells [18], which is the major place for CPT biosynthesis in *C. acuminata* according to a previous report [19]. Recently, a homolog of AtABCG40 from *Artemisia annua*, AaABCG40, was also identified as an ABA transporter, which demonstrates conserved function of ABCG40 homologs for transporting ABA in planta [23].

In the present study, we successfully isolated an *ABCG40* homolog (*CaABAT*) from *C. acuminata*, and sequence analysis showed that this identified *CaABAT* has high homology and contains the same conserved domain with AtABCG40 (Figure 1). For further understanding of the structure and possible function of CaABAT, a 3D model of CaABAT was constructed through subsection homology modeling using a Swiss-model server. The results showed that the crystal structure of ATP-binding cassette subfamily G member 2 (PDB code: 5nj3), as the template, shares 31.55% sequence identity and 36% similarity with 167–635 aa of CaABAT (as Model 1 in Appendix A). Moreover, ATP-binding cassette sub-family G member 5 (PDB code: 5do7) shares 32.13% sequence identity and 35% similarity with 857–1455 aa of CaABAT (as Model 2 in Appendix A). These templates were then artificially assembled into 3D models (Appendix A). ATP-binding cassette transporters are a highly conserved family of ATP-driven pump proteins consisting of two hydrophobic transmembrane domains (TMDs) and two cytosolic domains known as the nucleotide-binding domains (NBDs) or nucleotide-binding folds (NBFs). The two hydrophobic TMDs constitute a membrane-spanning pore (the red boxes), while the two cytosolic domains contain the ATP-binding motifs (the yellow box). The structure of this protein is consistent with a previous study [24]. Thus, we hypothesized that ABA may cause partial conformational changes in TMD and NBD by binding to the TMD of the transporter and activating NBD to release ADP.

### 3.2. Effect of CaABAT on ABA-Regulated Camptothecin Biosynthesis

Once abiotic stressors or developmental cues up-regulate the accumulation levels of endogenous ABA, PYR/PYL/RCAR (an ABA receptor) binds ABA and interacts with PP2C (a negative regulator) to inhibit protein phosphatase activity [25]. Then, SnRK2 (a positive regulator) is released from PP2C-dependent regulation and activated to phosphorylate downstream factors, such as the AREB/ABF bZIP-type transcription factor or membrane proteins involving ion channels [26]. After the phosphorylation of the bZIP transcription factor binding response element ABRE, the expression of ABA response genes was activated. Subsequently, ABA response was launched to resist external stimuli, such as the regulation of seed germination, the biosynthesis of secondary metabolites, stomatal closure, and drought resistance [26]. Although ABA transporters have been validated as signal cascade amplifiers in ABA signaling, efficient function of these transporters still depends on ABA accumulation levels in plants. In this work, since there was no excessive ABA accumulation in plants, only reduced accumulation levels of CPT could be observed in *CaABAT*-silenced leaves, while no variation in CPT accumulation levels could be found in *CaABAT*-overexpressing leaves. Therefore, appropriate regulation of ABA biosynthesis could be very important for ABA transporter-mediated promotion of CPT biosynthesis in *C. acuminata*.

By silencing the expression of CaABAT, four CPT biosynthesis pathway genes in the iridoid pathway could be obviously regulated, which demonstrates that the iridoid pathway should be the target of ABA signaling for regulating CPT biosynthesis in *C. acuminata*. Further, promoter sequence analysis showed that ABA responding element (ACGTG) could be found in *Ca7DLGT* (from −987 to −992) and *CaG8O* (from −263 to −268) genes (Appendix A), which additionally indicates the expression profile of these genes for responding to ABA signaling.

It is well known that ABA, as a signaling molecule, plays very important roles in plants, and it could be produced in response to stress, such as drought and low temperature. In *Artemisia annu*, it has been reported that the ABCG40 homolog AaABCG40 could enhance artemisinin accumulation and modulate drought tolerance [23]. In this work, we validated the importance of ABA signaling to CPT production mediated by the ABCG40 homolog, CaABAT. Considering that plants respond to most abiotic stresses through the ABA signaling pathway, translocation of endogenous ABA by ABA transporters, such as CaABAT, could be an important biological process for connecting the correlation of abiotic stresses and CPT biosynthesis in *C. acuminata*.

## 4. Materials and Methods

### 4.1. Plant Materials and Growth Conditions

*C. acuminata* seedlings were obtained by in vitro rapid propagation described in our previous report [22]. After 45 days of growth, seedlings were transferred to soil and grown at 25 °C with a 16 h photoperiod provided by cool white fluorescent light (40 μmol m^−2^s^−l^). After 4 weeks of growth, the new leaves were selected for the following experiments.

### 4.2. Isolation and Characterization of CaABAT

A homolog of AtABCG40 in *C. acuminata* was obtained by blast analysis in the *C. acuminata* transcriptome database (http://medicinalplantgenomics.msu.edu/, accessed on 8 March 2017), and then this ABA transporter was designated as *Camptotheca acuminata* ABA transporter (*CaABAT*). The full-length cDNA of *CaABAT* with *Kpn*I/*Sac*I at both ends was amplified by nested PCR using the following primers (Appendix A): first pair, *CaABAT*-FL-F and *CaABAT*-FL-R; second pair: *CaABAT*-*Kpn*I-F and *CaABAT*-*Sac*I-R. PCR products were then ligated into the pGEM-T easy vector (Promega, Beijing) to produce plasmid T-*CaABAT*. The *CaABAT* amino acid sequence and transmembrane structure were analyzed using MEGA6 and TMHMM 2.0, respectively. Then, conserved domains of *CaABAT* were analyzed using InterPro 91.0 [27,28].

### 4.3. RNA Extraction and Real-Time PCR Analysis

Plant tissues were ground into a fine powder within liquid nitrogen, and the total RNA was extracted using TRIzol reagent (Invitrogen, Shanghai, China). The RNA pellets were dissolved in DEPC-treated water, and the quality and quantity of total RNA were determined at OD 260 and 280 nm using a UV-2400 spectrophotometer (Shanghai Sunny Hengping Scientific Instrument Co., Ltd., Shanghai, China). After the extracted RNA was digested with DNase I, first-strand cDNA was synthesized using an M-MLV RTase cDNA Synthesis kit (TaKaRa Biotech Co., Ltd., Dalian, China) for real-time PCR analysis. The expression levels of target genes were analyzed using LightCycler 480 software (Roche) and normalized to *CaUBC* through the 2^–ΔΔCT^ method [29]. Primers were designed based on sequences from *CaUBC* and other target genes (Appendix A).

### 4.4. Transient Overexpression and Virus-Induced Gene Silencing (VIGS) of CaABAT in C. acuminata

For overexpression vector construction, full-length cDNA fragment of *CaABAT* was obtained from T-*CaABAT* digested by *Kpn*I/*Sac*I and cloned to pBIGD overexpression vector pre-digested with *Kpn*I/*Sac*I to produce pBIGD-*CaABAT* vector. Both empty vector pBIGD and pBIGD-*CaABAT* were then transformed into *Agrobacterium tumefaciens* strain GV3101 to treat *C. acuminata* leaves [22]. The plants were grown for 3 weeks until transformed leaves were harvested for further gene expression and CPT accumulation analysis. For constructing VIGS vectors, a 543 bp *CaABAT* fragment with *Kpn*I/*Sac*I restriction sites at both ends was PCR-amplified by using T-*CaABAT* plasmid as the template. The PCR product was then cloned into pGEM-T easy vector, and the *CaABAT* fragment was then obtained by *Kpn*I/*Sac*I double digestion. The *CaABAT* fragment was mobilized to the pTRV2 vector pre-digested with *Kpn*I/*Sac*I to produce pTRV2-*CaABAT*. The pTRV1, pTRV2, and pTRV2-*CaABAT* vectors were transformed into *Agrobacterium* strain GV3101, and VIGS experiments were carried out according to the method developed previously in the lab [30]. The plants were grown for 3 weeks until VIGS-treated leaves were harvested for further gene expression and CPT accumulation analysis.

### 4.5. Functional Identification of CaABAT in Yeast Cells

For yeast overexpression vector construction, pDRGAL vector was digested with *Not*I, followed by dephosphorylation of the digested backbone of pDRGAL. Full-length cDNA of *CaABAT* was obtained by *Not*I digestion of T-*CaABAT* plasmid and cloned to pre-treated pDRGAL vector mentioned above to produce pDRGAL-*CaABAT*. The pDRGAL and pDRGAL-*CaABAT* vectors were then transformed into the yeast strain AD12345678. The transformants containing pDRGAL and pDRGAL-*CaABAT* vectors were grown in SD/-URA medium until OD 600 nm = 0.8. A 1–4 µL amount of yeast cultures was then spotted on 1/2 SG/-URA medium plates containing 0, 100, 200, 300, or 500 µg/mL [20] of ABA, and the inhibitory effect of ABA on yeast growth was observed.

For further confirming ABA transport activity of CaABAT, transport assay in yeast cells was carried out. Yeast transformants containing pDRGAL and pDRGAL-*CaABAT* vectors were grown on SD/-URA medium until OD600 = 0.8. Yeast cells were then collected by centrifugation and re-suspended in 1/2 SG/-URA medium containing 100 µM of ABA with shaking at 180 rpm. Cells were harvested at the indicated time by centrifugation and lysed in a TissueLyser after being washed with distilled water three times. ABA accumulated inside yeast cells was then extracted in methanol, and ABA amounts were determined by HPLC analysis [22].

### 4.6. Subcellular Localization of CaABAT

The coding region of *GFP* (with stop codon) was PCR-amplified and cloned to pBIGD vector to produce pBIGD-*GFP*. For GFP-CaABAT fusion protein expression, both coding regions of *GFP* (without stop codon) and *CaABAT* were fused by PCR and then cloned to pBIGD vector to produce pBIGD-*GFP*-*CaABAT*. The constructed pBIGD-*GFP* and pBIGD-*GFP*-*CaABAT* vectors were transformed into *Agrobacterium tumefaciens* GV3101 to treat onion epidermal cells according to our previous report [20]. The *GFP* signal of the transformed cells was detected by laser scanning confocal microscopy (LSCM, Zeiss, Oberkochen, Germany).

### 4.7. Exogenous Plant Hormone Treatment

For evaluating the effect of exogenous plant hormones on pathway gene expression in *C. acuminata* leaves, seedlings were gently removed from soil, soaked in 100 µM ABA, 20 µM MeJA, or 20 µM SA for 0, 2, or 4 h. The leaves were harvested at the indicated time for target gene expression analysis. For analyzing the effect of ABA on CPT accumulation in *C. acuminata* leaves, seedlings were sprayed with 100 µM of ABA containing 0.1% DMSO (ABA was pre-dissolved in DMSO) for 0, 24, 48, or 72 h, while DMSO was used as negative control. Leaves were then harvested at the indicated time, and CPT contents were determined by HPLC analysis [22].

## Figures and Tables

**Figure 1 ijms-23-16120-f001:**
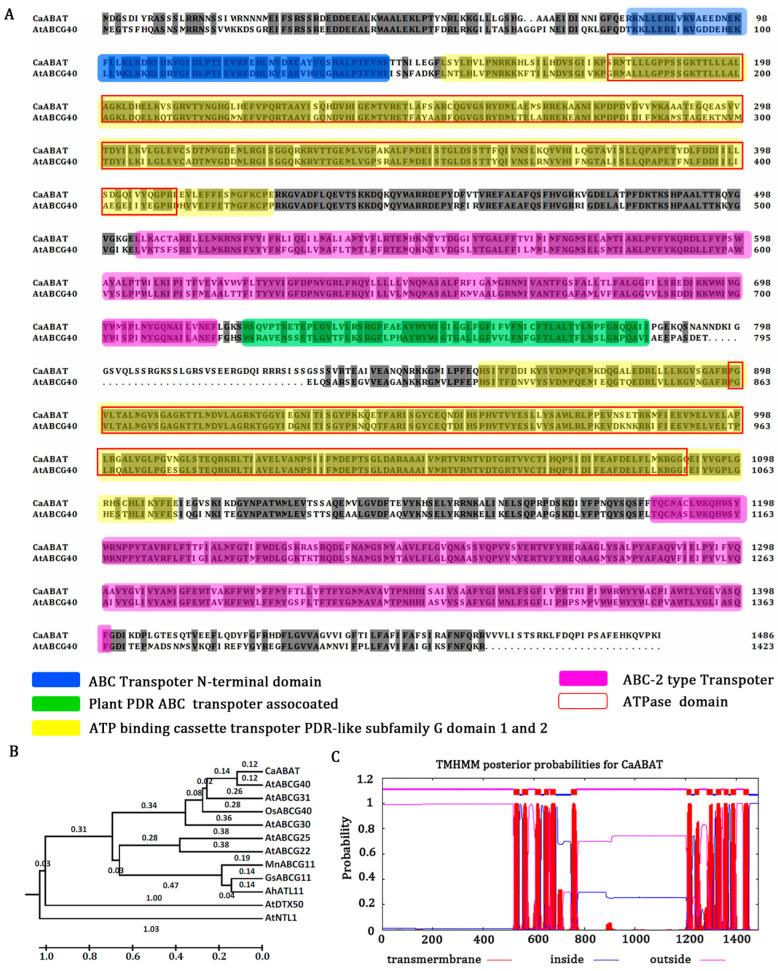
CaABAT is a member of the ABC transporter family. (**A**) Alignment of amino acid sequences of CaABAT with AtABCG40. ABC transporter domain (CaABAT 81–143 aa), ATP binding cassette transporter PDR-like subfamily G domain 1 (CaABAT 152–425 aa) and domain 2 (857–1112 aa), ABC-2 type transporter (504–716 aa, 1185–1399 aa), ABC transporter associated (CaABAT 721–784 aa), ATPase domain (CaABAT 177–409 aa, 897–1089 aa). (**B**) Phylogenetic relationships of CaABAT with other ABA transporters by MEGA6. AtABCG31: Q7PC88.1; AtABCG30: Q8GZ52.2; GsABCG11: KHN26506; MnABCG11: XP_010094142; AtABCG11: Q8RXN0.1; AtABCG40: Q9M9E1.1; AtABCG25: Q84TH5.1; AtNTL1: Q8H157.1; AtDTX50: Q9FJ87.1; AtABCG22: Q93YS4.1; AhATL11: AQW44869; CaABAT: AUB45106; OsABCG40: Q8GU85. (**C**) Transmembrane structure of CaABAT was analyzed using TMHMM 2.0 (online at http://www.cbs.dtu.dk/services/, accessed on 2 November 2017). CaABAT protein contains a total of six transmembrane structures. Transmembrane helical structures are represented by amino acid residues 520–542, 555–577, 605–627, 640–659, 664–686, 747–769, 1204–1223, 1236–1253, 1284–1306, 1319–1341, 1351–1370, 1377–1399, and 1428–1450 of the CaABAT protein. Consequently, residues 543–554, 628–639, 687–746, 1224–1235, 1307–1318, 1371–1376, and 1451–1487 are located inside the bilayer (i.e., are intracellular). Amino acid residues 1–519, 578–604, 660–663, 770–1203, 1254–1283, 1342–1350, and 1400–1427 are outside the bilayer.

**Figure 2 ijms-23-16120-f002:**
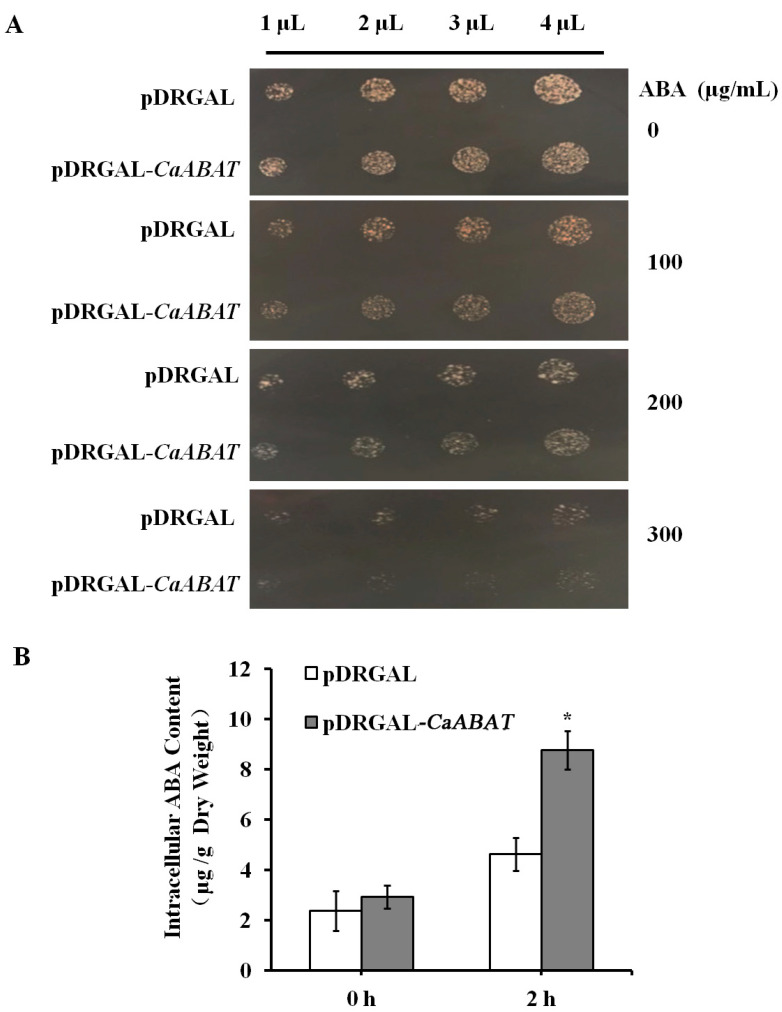
CaABAT is an ABA influx transporter. (**A**) The growth of *CaABAT* yeast transformants in a medium containing ABA. 1 to 4 μL of yeast cultures (OD_600_ = 0.1) were spotted on a 1/2 SG/-URA medium plate for 24 h at 28 °C. (**B**) ABA import activity of CaABAT. Yeast cells transformed with pDRGAL and pDRGAL-*CaABAT* were suspended in 1/2 SG/-URA medium supplemented with 100 µM ABA. The error bars represent standard deviations from three biological replicates, and asterisks indicate statistically significant differences compared with pDRGAL. * *p* < 0.05.

**Figure 3 ijms-23-16120-f003:**
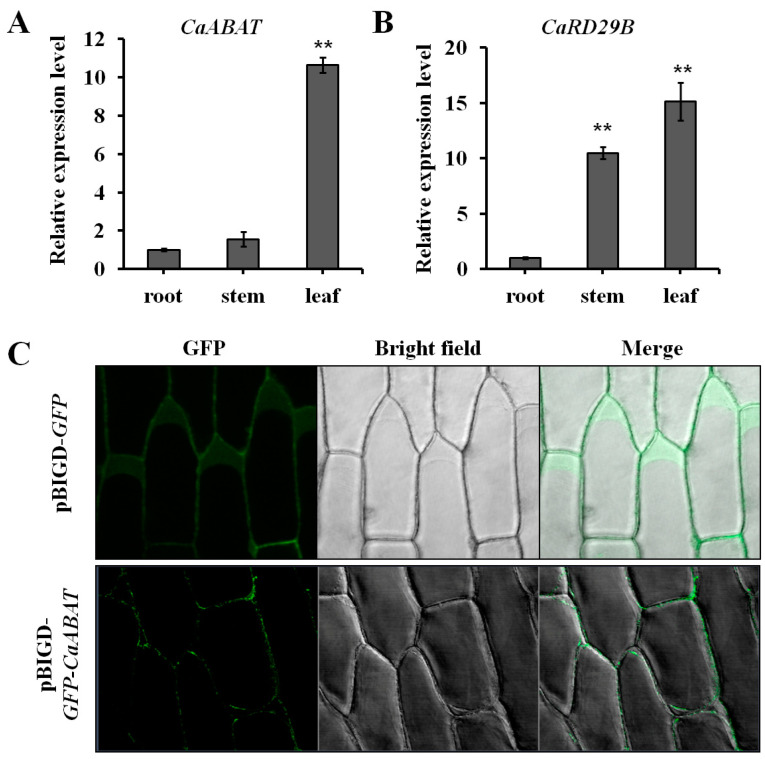
Expression analysis and subcellular localization of CaABAT. (**A**) Tissue-specific expression analysis of CaABAT in *C. acuminata*. (**B**) Tissue-specific expression analysis of CaRD29B in *C. acuminata*. (**C**) Subcellular localization of CaABAT in onion epidermal cells. The error bars represent standard deviations from three biological replicates, and asterisks indicate statistically significant differences compared with pDRGAL. ** *p* < 0.01.

**Figure 4 ijms-23-16120-f004:**
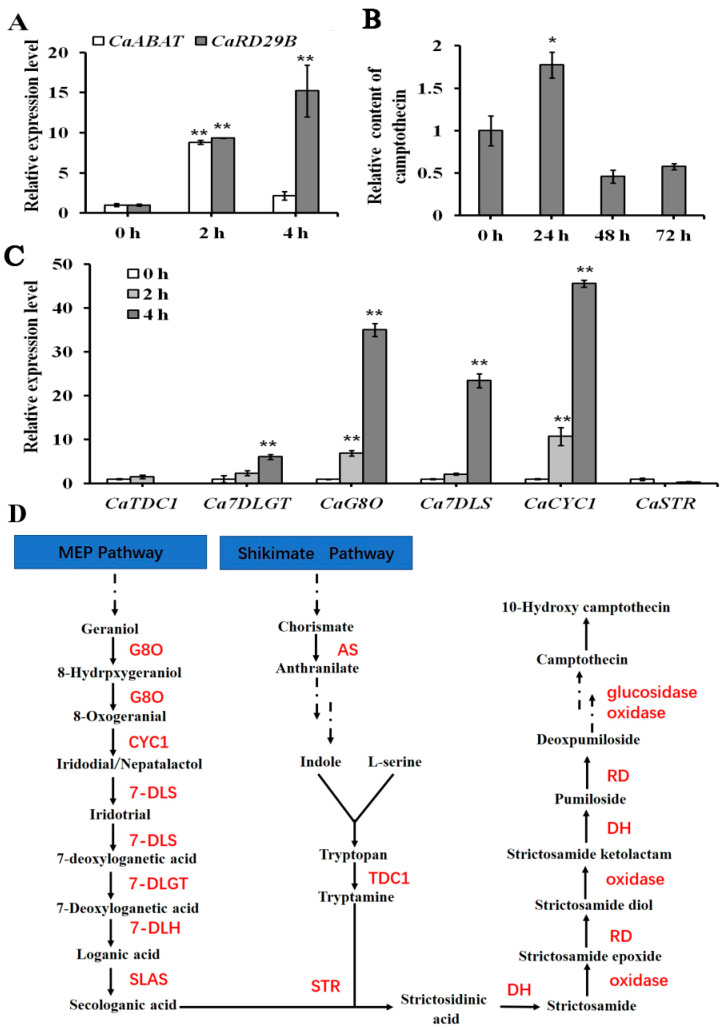
Gene expression and CPT accumulation analysis in response to exogenous ABA treatment in *C. acuminata* leaves. (**A**) Relative expression levels of *CaABAT* and *CaRD29B* in response to ABA treatments. (**B**) Accumulation levels of CPT in response to ABA treatments. (**C**) Relative expression levels of CPT biosynthesis genes in response to ABA treatments. The error bars represent standard deviations from three biological replicates, and asterisks indicate statistically significant differences compared with 0 h. * *p* < 0.05, ** *p* < 0.01. (**D**) Proposed CPT biosynthesis pathway in *C. acuminata* [22]. G8O, geraniol-8-oxidase; CYC1, Cyclase 1; 7DLS, 7-deoxyloganetic acid synthase; 7-DLGT, 7-deoxyloganetic acid glucosyltransferase; 7-DLH, 7-deoxyloganic acid hydroxylase; SLAS, secologanic acid synthase; AS, anthranilic acid synthetase; TDC, tryptophan decarboxylase; STR, strictosidinic acid synthase; DH, dehydration; RD, reduction.

**Figure 5 ijms-23-16120-f005:**
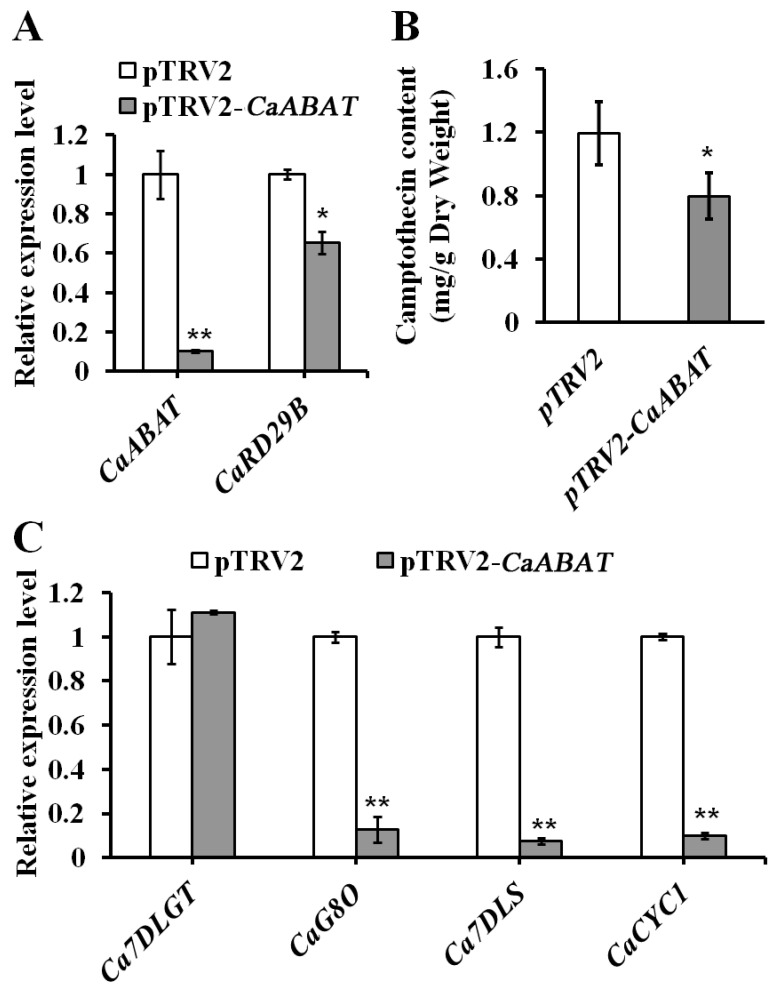
Virus-induced gene silencing (VIGS) of *CaABAT* in *C. acuminata* leaves. (**A**) Relative expression levels of *CaABAT* and *CaRD29B* in pTRV2 control and *CaABAT*-silenced leaves. (**B**) CPT accumulation levels in pTRV2 control and *CaABAT*-silenced leaves. (**C**) Relative expression levels of CPT biosynthesis genes in pTRV2 control and *CaABAT*-silenced leaves. The error bars represent standard deviations from three biological replicates, and asterisks indicate statistically significant differences compared with pTRV2 control. * *p* < 0.05, ** *p* < 0.01.

## Data Availability

Not applicable.

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
