# Peer review of "An ABCG-Type Transporter Facilitates ABA Influx and Regulates Camptothecin Biosynthesis in Camptotheca acuminata"

_ijms, 2022, doi:10.3390/ijms232416120_

Round 1

Reviewer 1 Report

In the manuscript ‘An ABCG-type transporter facilitates ABA influx and regulates camptothecin biosynthesis in Camptotheca acuminate (Manuscript ID: ijms-2063453)’, authors studied the function of one of the ABCB-type transporter from Camptotheca acuminate (CaABAT) that could function as ABA transporter. Although ABA transport potential is shown in yeast system, it would have been ideal to validate the function of CaABAT in planta as an ABA transporter. I do not know if grafting approach can be tried out using young plantlets of Camptotheca acuminate, considering its high levels in leaves compared to stem and root. Or even transient infiltration of ABAT overexpression construct, followed by grafting, and then, quantifying the levels of ABA in stem or root. I found it difficult to follow through the manuscript. Entire manuscript needs significant improvement for English language.  

In Figure 4, please provide a schematic of the camptothecin (CPT) biosynthesis pathway with the genes and enzymes involved in the pathway. This will be useful for better understanding of the results from Figures 4 and 5.

Line 91: Please provide citation for using UBC gene as a reference. Also, provide an appropriate citations for using 2^-deldelCT method for data analysis.  

Results section:

Lines 151-154 are M&M sentences, please delete them or move to their respective locations. The same is true for line 164.

Lines (209-222): Needs significant rephrasing.

Lines (205, 228, 263): ‘biological vector replicates’ should be written as ‘biological replicates (n)’.

Line 260: ‘real-time Q-PCR’ should be written as ‘real-time q-PCR’

Line 282: Please revise this sentence.

Line 303: Please delete the citation written in this sentence and cite number ‘32’ for this.

Line 306: Please delete the words ‘and the same conserved domain’.

Lines 322-324: Please revise this sentence.

Line 327: ‘abiotic stressers’ should be written as ‘abiotic stress conditions’

Line 342 and 349-350: Please revise these sentences.

Lines 349-363: Needs significant revising.

Line 355: Please delete the sentence ‘This also confirmed the above conclusion’.

Line 350: The word ‘regulated’ should be ‘differentially expressed’.

Line 361: The word ‘stresses’ should be written as ‘stress conditions’.

Author Response

Dear Reviewer,

Thanks for the comments related to our paper entitled “An ABCG-type transporter facilitates ABA influx and regulates camptothecin biosynthesis in Camptotheca acuminata”. We believe that we have made more revisions suggested by reviewer of the manuscript and that it has helped to significantly improve its value and meaning for broader audience. We hope that this revision will meet expectations and that might be acceptable for publication in International Journal of Molecular Sciences. We have attempted to highlight our responses and manuscript changes in blue to make it easier to see where changes have been made.

Thank you,

Yanyan Wang

PhD, Lecturer

School of Biological Engineering

Dalian Polytechnic University

Dalian,

Liaoning Province

China

Reviewer:

Comment 1:

In the manuscript ‘An ABCG-type transporter facilitates ABA influx and regulates camptothecin biosynthesis in Camptotheca acuminata (Manuscript ID: ijms-2063453)’, authors studied the function of one of the ABCB-type transporter from Camptotheca acuminata (CaABAT) that could function as ABA transporter. Although ABA transport potential is shown in yeast system, it would have been ideal to validate the function of CaABAT in planta as an ABA transporter. I do not know if grafting approach can be tried out using young plantlets of Camptotheca acuminata, considering its high levels in leaves compared to stem and root. Or even transient infiltration of ABAT overexpression construct, followed by grafting, and then, quantifying the levels of ABA in stem or root.

Response to comment 1: For identifying transport activity of certain plant specific transporters, yeast cells and X. laevis oocytes are generally used for the investigation according to their clean background related to substrates transport. Plants are not good system for testing direct transport activity of ABA, since many ABA transporters have already presented in plant cells and most of them are responding to ABA. The presence of these endogenous ABA transporters should extremely affect the results for examining ABA transport activity mediated by CaABAT alone. According to this consideration, we select yeast system for transport assay for confirming ABA transport activity of CaABAT.

Comment 2:

I found it difficult to follow through the manuscript. Entire manuscript needs significant improvement for English language.

Response to comment 2: Thank you very much for the corrections and we are truly sorry about our careless writing of the paper. We have corrected all mistakes suggested by reviewer and made improvement of the language.

Comment 3:

In Figure 4, please provide a schematic of the camptothecin (CPT) biosynthesis pathway with the genes and enzymes involved in the pathway. This will be useful for better understanding of the results from Figures 4 and 5.

Response to comment 3: Thank you very much for the suggestion and a schematic of CPT biosynthesis pathway has been added in the revised manuscript as Figure 4D.

Comment 4:

Line 91: Please provide citation for using UBC gene as a reference. Also, provide an appropriate citations for using 2–ΔΔCT method for data analysis.

Response to comment 4: Thanks for the suggestions, the references related to UBC gene and 2–ΔΔCT method have been added in the revised manuscript (line 79).

Comment 5:

Results section:

Lines 151-154 are M&M sentences, please delete them or move to their respective locations. The same is true for line 164.

Response to comment 5: Thanks for the suggestion, revision has been made, and highlighted in blue in the revised manuscript (lines 136-138, 146-147).

Comment 6:

Lines (209-222): Needs significant rephrasing.

Response to comment 6: Sorry about our careless writing and we have re-written the whole section and highlighted in blue in the revised manuscript (line 186-203).

Comment 7:

Lines (205, 228, 263): ‘biological vector replicates’ should be written as ‘biological replicates (n)’.

Response to comment 7: We have corrected the mistakes in the revised manuscript (lines 182, 207, 234).

Comment 8:

Line 260: ‘real-time Q-PCR’ should be written as ‘real-time q-PCR’

Response to comment 8: We have corrected this improper description in the revised manuscript.

Comment 9:

Line 282: Please revise this sentence.

Response to comment 9: We have re-written this sentence and highlighted in blue in the revised manuscript (line 251-259).

Comment 10:

Line 303: Please delete the citation written in this sentence and cite number ‘32’ for this.

Response to comment 10: Thanks very much and we have made the correction in the revised manuscript (line 277-279).

Comment 11:

Line 306: Please delete the words ‘and the same conserved domain’.

Response to comment 11: The description has been deleted in the revised manuscript.

Comment 12:

Lines 322-324: Please revise this sentence.

Response to comment 12: The sentence has been re-written according to text re-organization in the revised manuscript.

Comment 13:

Line 327: ‘abiotic stressers’ should be written as ‘abiotic stress conditions’

Response to comment 13: Thanks very much and the correction has been made in the revised manuscript.

Comment 14:

Line 342 and 349-350: Please revise these sentences.

Lines 349-363: Needs significant revising;

Line 355: Please delete the sentence ‘This also confirmed the above conclusion’.

Line 350: The word ‘regulated’ should be ‘differentially expressed’.

Line 361: The word ‘stresses’ should be written as ‘stress conditions’.

Response to comment 14: Thanks very much for the corrections and we have checked the whole manuscript and these inappropriate descriptions have been re-written in the revised manuscript (line 309-332).

Reviewer 2 Report

Dear authors,

The manuscript is well written with good presentation and  hight quality data.

I am just concerning about the plant phenotype. If the authors produce a plant that accumulates more ABA, if in one hand is good for Camptothecin  production, in other hand the plant can suffer the effect of mimic a hydric stress. I would like the authors present some images of the plants and additional information in the discussions.

Author Response

Dear Reviewer,

Thanks for the comments related to our paper entitled “An ABCG-type transporter facilitates ABA influx and regulates camptothecin biosynthesis in Camptotheca acuminata”. We believe that we have made more revisions suggested by reviewer of the manuscript and that it has helped to significantly improve its value and meaning for broader audience. We hope that this revision will meet expectations and that might be acceptable for publication in International Journal of Molecular Sciences. We have attempted to highlight our responses and manuscript changes in blue to make it easier to see where changes have been made.

Thank you,

Yanyan Wang

PhD, Lecturer

School of Biological Engineering

Dalian Polytechnic University

Dalian, Liaoning Province

China

Reviewer:

Comment 1:

The manuscript is well written with good presentation and high quality data.

I am just concerning about the plant phenotype. If the authors produce a plant that accumulates more ABA, if in one hand is good for Camptothecin production, in other hand the plant can suffer the effect of mimic a hydric stress. I would like the authors present some images of the plants and additional information in the discussions.

Response to comment 1: Thanks for reviewer’s suggestion and we re-organized discussion part and such concerns have been discussed in the revised manuscript (line 277-279).

Round 2

Reviewer 1 Report

Thank you for addressing my questions and revising the manuscript.